# The Wheat Annexin *TaAnn12* Plays Positive Roles in Plant Disease Resistance by Regulating the Accumulation of Reactive Oxygen Species and Callose

**DOI:** 10.3390/ijms242216381

**Published:** 2023-11-16

**Authors:** Beibei Shi, Weijian Liu, Qing Ma

**Affiliations:** 1Shaanxi Key Laboratory of Chinese Jujube, College of Life Sciences, Yan’an University, Yan’an 716000, China; shibeibei@yau.edu.cn (B.S.); 15550856083@163.com (W.L.); 2State Key Laboratory of Crop Stress Biology for Arid Areas, College of Plant Protection, Northwest A&F University, Yangling 712100, China

**Keywords:** annexins, wheat, plant immunity, ROS, callose

## Abstract

(1) Annexins are proteins that bind phospholipids and calcium ions in cell membranes and mediate signal transduction between Ca^2+^ and cell membranes. They play key roles in plant immunity. (2) In this study, virus mediated gene silencing and the heterologous overexpression of *TaAnn12* in *Arabidopsis thaliana* Col-0 trials were used to determine whether the wheat annexin *TaAnn12* plays a positive role in plant disease resistance. (3) During the incompatible interaction between wheat cv. Suwon 11 and the *Puccinia striiformis* f. sp. *tritici* (*Pst*) race CYR23, the expression of *TaAnn12* was significantly upregulated at 24 h post inoculation (hpi). Silencing *TaAnn12* in wheat enhanced the susceptibility to *Pst*. The salicylic acid hormone contents in the *TaAnn12*-silenced plants were significantly reduced. The overexpression of *TaAnn12* in *A. thaliana* significantly increased resistance to *Pseudomonas syringae* pv. *tomato* DC3000, and the symptoms of the wild-type plants were more serious than those of the transgenic plants; the amounts of bacteria were significantly lower than those in the control group, the accumulation of Reactive Oxygen Species (ROS)and callose deposition increased, and the expression of resistance-related genes (*AtPR1*, *AtPR2*, and *AtPR5*) significantly increased. (4) Our results suggest that wheat *TaAnn12* resisted the invasion of pathogens by inducing the production and accumulation of ROS and callose.

## 1. Introduction

Annexins, which bind phospholipids and calcium ions in cell membranes to mediate signal transduction between Ca^2+^ and cell membranes, are ubiquitous in plants, animals, and fungi [1]. After regulation via a range of local environments (e.g., Ca^2+^, pH, voltage, and lipids) or atypical ordering motifs, plant annexins work by inserting themselves into cytosolic membranes and are Ca^2+^-dependent members of the phospholipid-binding protein family, the functions of which include exocytosis, peroxidase, and ATPase/GTPase activities; binding actin; regulating Ca^2+^ channel activity; and transportation [2]. Annexins have been shown to be involved in H_2_O_2_-activated Ca^2+^ fluxes in plants [3,4], act as Ca^2+^-permeable transporters, and exhibit peroxidase activity [5]. Consequently, annexins link Ca^2+^, redox, and lipid signaling to coordinate the development of responses to both biotic [6,7,8] and abiotic stresses [9,10,11,12,13].

In response to abiotic stresses, such as NaCl stress [13], PEG stress (simulated drought) [11], abscisic acid (ABA) stress [12], and high [10] or low temperatures [9], the expression levels of annexins are regulated, illustrating the involvement of annexins in plant stress responses. *Arabidopsis thaliana* MYB30 regulates high-temperature stress responses via annexin-mediated calcium signaling, which binds MYB30 to the promoters of AnnAt1 and AnnAt4 to inhibit the expression of annexins [14]. Under abiotic stresses, OsANN1 overexpression in plants promoted the activities of superoxide dismutase (SOD) and catalase (CAT) to regulate the H_2_O_2_ content, indicating that OsANN1 is involved in a feedback mechanism in H_2_O_2_ production, whereas OsANN1-knockdown plants were susceptible to heat and drought stresses [15]. The chitin receptor CHITIN ELICITOR RECEPTOR KINASE 1 (CERK1) interacted with ANNEXIN 1 to function in chitin or salt signaling in [16]. 

The functions of annexins in plant immunity have been identified [6,7,8]. The heterologous expression of AnnBj1 (*Brassica juncea*) in tobacco provides tolerance to *Phytophthora parasitica* var. *nicotianae* [17]. Similarly, the CkANN of *Cynanchum komarovii* enhanced resistance to *Fusarium oxysporum* in transgenic cotton in [18]. Additionally, after powdery mildew infection, the expression of *A. thaliana* AnnAt1 increased significantly [19]. However, AtAnn8 negatively regulated the RPW8.1-mediated resistance against powdery mildew and cell death [20]. In another study, due to their Ca^2+^-binding domains, annexins reduced the activity of Ca^2+^ on callose ((1→3)-β-glucan) synthase, and cotton annexin inhibited the activities of callose synthase [21]. The callose deposition increased the CmbHLH18 resistance to the necrotrophic fungus *Alternaria brassicicola*, which kills epidermal cells by secreting toxic metabolites and proteins [22], indicating that the accumulation of callose affects plants’ defense against biotic stress [23]. Regarding nematode resistance, the MIF-like effector MiMIF-2 of *Meloidogyne incognita* protected nematodes from oxidative stress by regulating the function of the plant annexins AnnAt1 and AnnAt4, and plants with AnnAt1 or AnnAt4 overexpression were more resistant to *M. incognita*, as illustrated by a reduced number of galls and nematodes inside the roots of the plants [24]. Hs4F01, a homologous protein of Arabidopsis AnnAt1, is an annexin-like effector that mimics plant annexins and regulates the defense response by interacting with 2OG-Fe(II) oxygenase (DMR6) [25].

An analysis of the wheat genome by Xu et al. [26] led to the identification of 25 putative annexin genes, which were assigned to subgenomes A, B, and D. Further analyses led to the finding that each haploid genome of wheat has 12 annexin gene family members (*TaAnn1-12*). Based on these sequences, increases in the expression of the annexin gene family in response to abiotic stresses were detected. Among them, the expression of *TaAnn12* was significantly induced by salt, drought, cold, and ABA [26]. However, it is unclear whether *TaAnn12* is involved in plant defense responses, especially in wheat resistant to stripe rust. Wheat (*Triticum* spp.) is the second largest major crop in the world, with wheat planting areas occupying approximately 220 million hectares worldwide [27]. Wheat stripe rust, caused by *Puccinia striiformis* f. sp. *tritici* (*Pst*), is one of the main fungal diseases affecting wheat. Due to the changing climate and environment, the races of wheat stripe rust have rapidly mutated, overcoming the resistance of wheat varieties currently protected by resistance genes [28]. Therefore, finding new ways to carry out studies on wheat disease resistance and cultivating wheat varieties with relatively durable resistance will provide new development directions for the genetic breeding of wheat disease resistance. In recent years, combining molecular breeding with phenotypic selection has provided great hope for the genetic diversity of stripe-rust-resistant wheat [27]. In this study, to analyze whether the wheat annexin *TaAnn12* plays a role in wheat defense, we detected the accumulation of *TaAnn12* in wheat leaves upon *Pst* infection, which was significantly upregulated during the incompatible interaction between wheat and the *Pst* race CYR23. Furthermore, barley stripe mosaic virus (BSMV)-induced gene silencing technology and heterologous transgenic overexpression in Arabidopsis assays were used to identify the function of *TaAnn12* in plant disease resistance. The results of this study provide important information for understanding the different functions of plant annexins. 

## 2. Results

### 2.1. The Expression of TaAnn12 Is Differently Induced during Pst Infection, Hormone Treatments, and Abiotic Stresses

Full-length primers were designed according to the known *TaAnn12* sequence in the NCBI sequence database (GenBank accession number: AK331881.1) to obtain a targeting gene from wheat cv. Suwon 11 cDNA using PCR technology, with the ORF sequence length being 945 bp. To analyze the function of *TaAnn12* in wheat defense, we detected the accumulation of *TaAnn12* in wheat leaves upon *Pst* infection. As shown in Figure 1a, in the incompatible interaction, compared with samples at 0 hpi, the expression of *TaAnn12* was significantly induced at 24 hpi, with a 6.3-fold increase, indicating that *TaAnn12* provides tolerance to biotic stress. 

Additionally, we analyzed how *TaAnn12* responded to an exogenous hormone stimulus. Different hormones were sprayed on wheat, and the results are shown in Figure 1b. The expression of *TaAnn12* mRNA increased significantly under the treatments of salicylic acid (SA), jasmonate (JA), and abscisic acid (ABA), suggesting that *TaAnn12* may be involved in hormone signaling pathways. The expression of *TaAnn12* responded to abiotic stresses (i.e., salt, drought, and cold) [26], and we also confirmed that the expression of *TaAnn12* was significantly induced under salt, drought, cold, and heat treatments (Appendix A). In addition, we found that the *TaAnn12* mRNA accumulations were different in roots, stems, and leaves (Appendix A).

### 2.2. Suppression of TaAnn12 Reduces Wheat Resistance to Pst

To examine the potential role of *TaAnn12* in the process of wheat stripe rust infection, we used barley stripe mosaic virus (BSMV)-induced gene silencing technology to silence *TaAnn12* during *Pst* infection. A specific 132 bp segment of *TaAnn12* was inserted into the BSMV: γ vector. Ten days after inoculating with a recombinant virus (BSMV: γ, BSMV: *TaPDS*, and BSMV: *TaAnn12*), obvious photo-bleaching (BSMV: *TaPDS*) and virus mosaic symptoms (BSMV: γ and BSMV: *TaAnn12*) were observed on fourth Su11 leaves (Figure 2a). Then, the new fourth leaves were inoculated with the *Pst* races CYR23 and CYR31, with the infected wheat leaves being observed, photographed, and recorded at 15 dpi. Remarkably, as shown in Figure 2b, typical hypersensitive reactions (HRs) appeared on the mock and BSMV: γ leaves after infecting with CYR23 (incompatible interaction), while the necrosis area in *TaAnn12*-silenced plants increased, and small amounts of uredia were produced. On the other hand, on wheat leaves inoculated with CYR31 (compatible interaction), *TaAnn12*-silenced plants produced more uredia compared with the control plants (Figure 2c). In addition, the silencing efficiency of *TaAnn12* was confirmed via RT-qPCR. Compared with BSMV: γ, the expression of *TaAnn12* reached 61–78% when inoculated with the *Pst* race CYR23, while it reached 54–70% when inoculated with the *Pst* race CYR31 in the *TaAnn12*-knockdown plants at 0, 24, 48, and 120 hpi (Figure 2d). 

To determine whether the resistance-related genes in *TaAnn12*-silenced plants were affected, the expression of the SA-pathway-related gene (*TaNPR1*), H_2_O_2_-pathway-related genes (*TaCAT* and *TaSOD*) and resistance-related genes (*TaPR1* and *TaPR2*) was quantified with RT-qPCR. The transcript levels of *TaNPR1*, *TaPR1*, and *TaPR2* were significantly reduced in the BSMV: *TaAnn12* leaves compared to BSMV: γ after inoculating with CYR23, which indicated that *TaAnn12* may affect wheat resistance to *Pst* by involving the SA signaling pathway. In contrast, the expression levels of *TaCAT* and *TaSOD* were found to be significantly upregulated in *TaAnn12*-gene-silenced plants (Figure 2e). 

Additionally, histological observations (i.e., branches, hyphal length, and colony area) were made on BSMV: *TaAnn12* leaves at 24 h, 48 h, and 120 h after inoculating with CYR23 to further clarify the function of *TaAnn12* in wheat resistance to stripe rust infection. The growth of *Pst* was expanded in BSMV: *TaAnn12* plants under a fluorescence microscope (Figure 3a). The hyphal length of gene-silenced plants was significantly increased compared with the control at 48 hpi. And the hyphal length and colony area of infected BSMV: *TaAnn12* plants increased significantly at 120 hpi (Figure 3b–d). In summary, these results demonstrated that *Pst* infection was enhanced in gene-silenced wheat, suggesting that *TaAnn12* plays an important role in attenuating stripe rust.

### 2.3. Reduced Accumulation of SA in TaAnn12-Silenced Plants

SA and JA are indispensable hormones in plant immune response signaling pathways [29]. The exogenous hormone treatment experiment showed that MeJA and SA induced the expression of *TaAnn12* (Figure 1b). In addition, the BSMV-VIGS system showed that the expression of the SA signaling pathway marker genes *TaNPR1*, *TaPR1*, and *TaPR2* were significantly downregulated in BSMV: *TaAnn12* plants (Figure 2e). In order to confirm the specific signaling pathway affecting *TaAnn12* under *Pst* infection, we detected (using liquid chromatography–mass spectrometry (LC/MS)) the accumulation of SA and JA in BSMV: *TaAnn12* and BSMV: γ at 18 h and 24 h after inoculating with the *Pst* race CYR23. As shown in Figure 4, the SA content of *TaAnn12*-silenced plants decreased by 24% compared to the control and was significantly decreased at 24 hpi (Figure 4a), while the JA concentration had no significant change (Figure 4b). These results indicated that *TaAnn12* regulates wheat resistance to stripe rust via the SA signaling pathway.

### 2.4. Overexpressing TaAnn12 in Arabidopsis Enhances Plant Defense

A transgenic overexpression of *TaAnn12* in *Arabidopsis thaliana* Col-0 materials was created to investigate the results obtained in the transient gene silencing experiments of *TaAnn12* in disease resistance, and we screened two T1-generation lines (*TaAnn12*-OE1 and -OE2). After the expansion and PCR identification of the T2 generation (*TaAnn12*-T2-OE1: 1, 4–7, 9, 10, 12, 13, and 15–18; *TaAnn12*-T2-OE2: 1–7, 8–11, and 13) (Appendix A), the T3-generation lines were injected with *Pseudomonas syringae* pv. *tomato* (*Pto*) DC3000, and the symptoms were observed at 4 dpi. The yellowing and wilting of wild-type plants were more serious than in transgenic plants (Figure 5a). Incidentally, the amounts of *Pto* DC3000 were isolated and detected, with the colony densities of *TaAnn12*-T3-OE1 and *TaAnn12*-T3-OE2 being significantly lower than Col-0 at 24 h (10^5.08^ CFU/cm^2^ and 10^5.08^ CFU/cm^2^) and 48 h (10^5.73^ CFU/cm^2^ and 10^5.40^ CFU/cm^2^) (Figure 5b). In addition, trypan blue was used to stain the necrotic area of *Pto* DC3000 at 24 hpi and 48 hpi, and the plants overexpressing *TaAnn12* showed more intense cell necrosis (Figure 5c). These results indicated that *TaAnn12*-overexpressing Arabidopsis plants were more resistant to *Pto* DC3000 than wild-type plants and that *TaAnn12* had a positive regulatory role in the process of resistance to *Pto* DC3000. 

Subsequently, the expression of defense-related genes was detected via RT-qPCR. Samples were taken at 0, 3, 8, and 24 h after inoculating with *Pto* DC3000, and the expression levels of *AtPR1*, *AtPR2*, *AtPR5*, and *AtPDF1.2* were detected. The results were as follows (Figure 6): compared with Col-0, the expression of *AtPR1*, *AtPR2*, and *AtPR5* in *TaAnn12*-T3-OE1 and *TaAnn12*-T3-OE2 plants was significantly increased at 8 and 24 hpi, while the JA signaling pathway marker gene *AtPDF1.2* had no significant change, which further confirmed that the overexpression of *TaAnn12* significantly increased the resistance of Arabidopsis to *Pto* DC3000.

### 2.5. TaAnn12 Affects Plant Resistance by Regulating ROS and Callose Accumulation

Reactive oxygen species (ROS; e.g., O^2−^, H_2_O_2_, OH•, and O_2_) signaling is related to plant immunity [30]. Herein, nitro-blue tetrazolium (NBT) and 3,3-diaminobenzidine (DAB) staining were used as indicators of the superoxide anion (O^2−^) and hydrogen peroxide (H_2_O_2_) to detect the ROS in Arabidopsis leaves inoculated with *Pto* DC3000. The NBT staining showed that Col-0 leaves had a small number of blue spots, while the *TaAnn12*-T3-OE1 and *TaAnn12*-T3-OE2 plants showed navy blue at 3 and 8 hpi, indicating that the overexpression of *TaAnn12* promoted O^2−^ accumulation (Figure 7a). Similarly, H_2_O_2_ accumulation was higher in overexpressing plants (Figure 7b). These results suggest that *TaAnn12* may affect plant immunity by regulating the production of ROS. 

Plant annexins are involved in the regulation of callose synthesis [21]. To determine whether *TaAnn12* affects callose accumulation, *TaAnn12*-T3-OEs were stained with aniline blue in Arabidopsis leaves inoculated with *Pto* DC3000. Fluorescent signals were generated and observed under a fluorescence microscope (Figure 8a). The callose accumulation of *TaAnn12*-T3-OE1 and *TaAnn12*-T3-OE2 was increased compared to the control, and the amount of callose deposition at 24 hpi was higher than at 12 hpi, indicating that overexpression of *TaAnn12* promotes the accumulation of callose and that the accumulation of callose gradually increases with time (Figure 8b).

## 3. Discussion

Annexins respond to biotic and abiotic stresses and are involved in signal transduction between calcium ions and cell membranes. They link Ca^2+^ and lipid signaling to participate in callose synthesis and ion transport [1]. According to the bioinformatics analysis of the wheat annexin gene family and its responses to abiotic stress, *TaAnn12* is significantly induced by salt, drought, cold, and ABA and plays key roles in responding to abiotic stresses. However, it has been unclear whether *TaAnn12* is involved in plant defense responses, especially in wheat resistant to stripe rust. In this study, we found that *TaAnn12* played positive roles in plant disease resistance.

Annexins have been identified in numerous studies of plant defenses [7,8]. Overexpression of AnnBj1 in tobacco improves resistance to *Phytophthora parasitica* var. *nicotianae* [17]. AnnAt1- or AnnAt4-overexpressing plants were more resistant to *M. incognita* [24]. *AtAnn1* and *AtAnn2* double-mutant *ann1ann2* showed more susceptibility to *Botrytis cinerea* [31], implicating the positive regulation effect of *AtAnn1* and *AtAnn2* in plant resistance against *B. cinerea*. Furthermore, the expression of AnnAt1 increased significantly during osmotic stress and powdery mildew infection [19,32,33]. In addition, *AtAnn8* was a negative regulator of RPW8.1-mediated resistance to powdery mildew and cell death [20]. In order to determine whether *TaAnn12* is involved in the interaction between plants and pathogens, in this study the expression levels of *TaAnn12* in stripe-rust-infected wheat were measured. Among them, in the incompatible interaction, TaAnn12 was upregulated at 24 hpi, indicating that *TaAnn12* positively responded to *Pst* CYR23, which was further improved by BSMV-VIGS technology. Afterwards, the *TaAnn12* gene was heterologously overexpressed in Arabidopsis. The transgenic plants enhanced *A. thaliana* defense and further clarified that *TaAnn12* played a positive regulatory role in disease resistance. 

The incompatible interaction between a plant and a pathogen causes a hypersensitive reaction (HR) to limit the growth of the pathogen [34], which is a local defense, caused primarily by the production and accumulation of ROS [35]. Overexpression of OsANN1 promoted SOD and CAT activity to regulate the accumulation of H_2_O_2_ [15]. In this study, the expression of *TaCAT* and *TaSOD*, which are H_2_O_2_-pathway-related genes, was increased in *TaAnn12*-silenced plants, accelerating the removal of ROS and leading to a decrease in the disease resistance level of silent plants. In addition, the accumulation areas of superoxide (O^2−^) and hydrogen peroxide (H_2_O_2_) in *TaAnn12* transgenic Arabidopsis were larger than in the control, and the necrotic area increased, indicating that *TaAnn12* induced the HR and the accumulation of ROS to exert disease resistance.

The accumulation of callose affects plant defense against biotic stress [23], and plant annexins are involved in regulating callose ((1→3)-β-glucan) synthase synthesis [21]. Oomycete annexin stimulated the (1→3)-β-d-glucan synthase activator [36], suggesting that specific annexins may have antagonistic effects in regulating callose enzyme activity [1]. To determine the functions of *TaAnn12* in callose accumulation, aniline blue staining was used to detect the callose content after inoculation with *Pto* DC3000, and the callose accumulation of *TaAnn12*-overexpressing Arabidopsis was increased compared to the control, showing that *TaAnn12* was involved in the accumulation of callose.

Plants have evolved complex signaling and defense mechanisms to defend against pathogen infections. Fungi have diverse lifestyles in which they deploy distinct strategies to interact with their host plants, including necrotrophic, biotrophic, and hemibiotrophic strategies [22]. For instance, the defense strategies of plants produce different stress signaling based on the type of invading pathogen (e.g., fungi or bacteria) or pathogenic lifestyle (biotroph, hemibiotroph, or necrotroph). The induction of defense genes is orchestrated by signaling networks that are directed by plant hormones. Salicylic acid (SA) and jasmonic acid (JA) are the major players, and SA plays an important role in plant responses to biotrophs and hemibiotrophs [29]. After a pathogen infects a plant, the host recognizes signaling generated by the pathogen to initiate a defense response in the plant, which promotes the synthesis of SA and activates the expression of downstream disease-associated protein (PR) genes, so that the plant can prevent the pathogen infection [37]. The CkANN enhanced the tolerance of transgenic cotton to *Fusarium oxysporum*, and the transcription level of *PRs* in CkANN transgenic cotton increased, indicating that CkANN is involved in PR protein-mediated SA-dependent defense responses [18]. In order to clarify the signaling pathway of *TaAnn12* in wheat resistance to stripe rust, wheat was treated with the exogenous hormones SA, MeJA, ET, and ABA, all of which induced the expression of *TaAnn12*. Furthermore, we measured the contents of SA and JA in *TaAnn12*-silenced wheat, and the results showed that the SA content in *TaAnn12*-silenced plants decreased significantly compared with the control, indicating that *TaAnn12* participated in the SA signaling pathway of plant immunity. On the other hand, after infection with *Pto* DC3000 in *TaAnn12*-overexpressing Arabidopsis, the expression of the defense-related genes *AtPR1*, *AtPR2*, and *AtPR5* was significantly increased, while the JA signaling pathway marker gene *AtPDF1.2* did not change significantly, further indicating that *TaAnn12* was involved in the SA signaling pathway. *TaAnn12* plays a positive role in regulating plant disease resistance through the SA signaling pathway.

## 4. Materials and Methods

### 4.1. Plants, Pathogen Material, and Treatment

Wheat (*Triticum aestivum*) cv. Suwon 11 (Su11) and Mingxian 169 were used. Fresh urediospores of *Puccinia striiformis* f. sp. *tritici* (*Pst*) were propagated on the Mingxian 169. The *Pst* race CYR23 had an incompatible interaction with Su11, triggering a typical hypersensitive reaction (HR) after inoculation, and the *Pst* race CYR31 had a compatible interaction with Su11, during which a large number of uredia were attached. Wheat seedlings were grown at 23 °C, with day and night periods of 16 h and 8 h, respectively. After inoculating with fresh urediospores of *Pst*, the wheat was placed in a dark incubator with a temperature of 10 °C and a humidity of 90% for 24 h and then placed in a 16 °C incubator (day/night cycle of 16 h/8 h) [38]. 

Wild-type *Arabidopsis thaliana* Columbia-0 (Col-0) was used as a transgenic Arabidopsis. Col-0 seeds were sterilized in 75% ethanol for 5 min and then immediately washed with sterile water three times. After surface sterilization, seeds were cultivated and germinated in a 1/2 MS medium (pH 5.5) and cultured at 4 °C for 2 days. After the emergence of the cotyledon, seedlings were transplanted into vermiculite seedling trays with a temperature of 23 °C and a day/night cycle of 10/14 h. *Pseudomonas syringae* pv. *tomato* (*Pto*) DC3000 was grown on King’s B (KB) medium with 25 mg/L rifampicin at 28 °C. Leaves of 4-week-old Arabidopsis were injected with *Pto* using a syringe (without a needle) [39]. 

The *Pto* DC3000 was grown on King’s B (KB) medium at 28 °C, and resuspended in 10 mM MgCl_2_ to OD600 = 0.002. Leaves of 4-week-old plants were infected with the bacterial suspension by pressing a 1 mL syringe (without a needle) against the abaxial side of the leaves and forcing the suspension through the stomata into the intercellular spaces, as described in the previous study [40]. Plants were sampled at 0, 3, 8, and 12 h and then used to extract RNA.

Ten-day-old Su11 plants were inoculated with the Pst races CYR23 and CYR31 and collected at 0, 12, 24, 36, 48, 72, 96, and 120 hpi. Ten-day-old Su11 plants were sprayed with 2 mM salicylic acid (SA), 100 µM abscisic acid (ABA), 100 µM ethylene (ET), and 100 µM methyl jasmonate (MeJA) and collected at 0, 0.5, 3, 6, 12, and 24 hpt. Ten-day-old Su11 plants were sprayed with 200 mM NaCl and 20% (*w*/*v*) PEG6000 and were placed in 4 °C and 37 °C incubators. Leaves were collected at 0, 1, 3, 6, 12, and 24 hpt [41]. 

### 4.2. Real-Time Quantitative PCR (RT-qPCR) Analysis and Statistical Analysis

The specific primers TaAnn12-Q-F/TaAnn12-Q-R were used for RT-qPCR, and *TaEF-1a* was used as the control gene (Appendix A). Total RNA was extracted according to the total RNA extraction kit (Invitrogen, Carlsbad, CA, USA). First-strand cDNA was prepared according to the FastKing RT Kit (Tiangen, Beijing, WI, China). The RT-qPCR reaction system and procedures referred to the ChamQTM SYBR qPCR Master Mix kit (Vazyme, Nanjing, China). The *TaEF-1a* gene (GenBank accession number: M90077.1) was used as the control gene of wheat, and the *AteIF4A* gene was used as the control gene of Arabidopsis [39]. The 2^−ΔΔt^ method was used to calculate the relative expression of genes. Each sample had three independent biological replicates. SPSS 23.0 was used to determine the statistical significance (*p* < 0.05) [42].

### 4.3. Barley Stripe Mosaic Virus (BSMV)-Mediated Gene Silencing

A specific fragment of *TaAnn12* was cloned into the BSMV: γ vector, which was used in SNG-VIGS (https://vigs.solgenomics.net/, accessed on 6 September 2021) to determine the specificity. After linearizing the BSMV: α, BSMV: β, BSMV: γ, BSMV: γ-*TaAnn12*, and BSMV: γ-*TaPDS* plasmid vectors with restriction endonuclease, a RiboMAX^TM^ Large Scale RNA Production Systems-T7 Kit (Promega, Austin, TX, USA) was used to reverse-transcribe BSMV RNAs in vitro. The BSMV: α, β, and γ genes were mixed proportionally with FES buffer and inoculated by rubbing leaves on the second leaf of a ten-day-old Su11 plant, where FES buffer was used as a blank control, BSMV:γ was used as a negative control, and BSMV: γ-*TaPDS* was used as a positive control. The sample was placed in a dark incubator with a temperature of 25 °C and a humidity of 90% for 24 h and then placed in a 25 °C incubator (day/night cycle of 16 h/8 h) [43]. Ten days after inoculation, photo-bleaching and mild chlorotic mosaic symptoms were detected on the fourth leaves of wheat plants inoculated with the *Pst* races CYR23 and CYR31, respectively. The phenotypes were observed and photographed at 15 dpi. Samples were taken at 0, 24, 48, and 120 hpi to extract RNA and at 24, 48, and 120 hpi for histological observation [41]. The expression level of *TaAnn12* in the leaves with BSMV:γ at each time point was standardized as 1. To observe *Pst* development, wheat leaves were decolorized with 100% ethanol and then stained with WGA AF488. They were observed under a NIKON-80i fluorescence microscope, with 30 infection points being calculated at each time point, including the number of hyphal branches, hyphal length, and colony area [44]. SPSS 23.0 was used to determine the statistical significance (*p* < 0.05). 

### 4.4. Detection of the Accumulation of Plant Hormones in TaAnn12-Silenced Plants

First, 200 mg of BSMV: γ and BSMV: γ-*TaAnn12* wheat leaves were sampled at 18 and 24 h after inoculating with CYR23. After grinding with liquid nitrogen, 1 mL of a precooled extraction liquid (methanol/water/glacial acetic acid = 90:9:1) was added and centrifuged to collect the supernatant. After repeating twice, samples were dried in nitrogen, and 200 µL of methanol was added to dissolve the samples. In order to detect the contents of salicylic acid (SA) and jasmonic acid (JA), SA and JA chemical molecular standard samples (Sigma, Shanghai, China) were prepared according to the concentration gradient. The standard samples and a sample of wheat leaves were tested using liquid chromatography–tandem mass spectrometry (LC/MS). Based on the standard curve, the accumulations of SA and JA were analyzed [41].

### 4.5. Production of Transgenic Arabidopsis

The full-length ORF of *TaAnn12* was constructed in a pCMBIA3301 (controlled by CaMV 35S promoter) vector, which was transformed into the *Agrobacterium tumefaciens* strain GV3101. Using a vacuum infiltration method, the overexpression vector was transformed into Col-0. Transgenic T1-generation seeds were selected with Kanamycin on a 1/2 MS medium. The seedlings were grown until they had four dark-green leaves and were transplanted into vermiculite seedling trays. After amplifying the insertion of the transgene in the genomic DNA using a PCR assay, transgenic lines were obtained in the T2 generation, and the T3-generation plants were tested for disease resistance [39].

### 4.6. Identification of Disease Resistance in Transgenic TaAnn12 Arabidopsis

*Pseudomonas syringae* pv. *tomato* (*Pto*) DC3000 was resuspended in 10 mM MgCl_2_ to OD_600_ = 0.002 (approximately 1 × 10^6^ CFU/mL). Leaf phenotypes were observed at 4 dpi, and samples were collected at 0, 3, 8, and 24 hpi for RNA extraction. To isolate and detect the amount of *Pto* DC3000, leaf samples were taken at 0, 24, and 48 hpi with a punch (diameter of 5 mm) and 0.1 mL of sterile water was added. After gradient dilution, single colonies were counted on KB medium with 25 mg/L rifampicin. For 3,3-diaminobenzidine (DAB) staining, samples were taken at 3, 8, and 24 hpi and treated with 1 mg/mL DAB (pH 3.8) for 30 min, which decolorized after dark staining for 8 h. The samples at each timepoint (3, 8, and 24 hpi) were stained using 1 mg/mL nitro-blue tetrazolium (NBT) for 30 min, which stained in the dark for 2 h before decolorization [45]. For trypan blue staining, samples were taken at 24 and 48 hpi and were boiled with a trypan blue solution for 3 min. Samples were collected at 12 and 24 hpi for callose staining. After decolorization, leaves were pretreated with 10% KOH and stained with 0.01% aniline blue in 67 mM K_2_HPO_4_ for 6 h [46].

## 5. Conclusions

In conclusion, our results revealed that wheat *TaAnn12* is a positive regulator of plant immunity that resists the invasion of a pathogen by inducing the production and accumulation of ROS and callose and regulates the disease resistance response of wheat through the SA signaling pathway. Thus, the expression of downstream disease resistance proteins is activated.

## Figures and Tables

**Figure 1 ijms-24-16381-f001:**
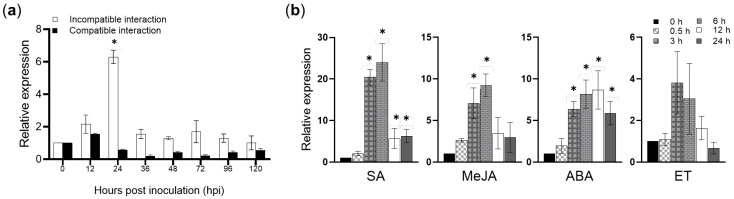
Real-time quantitative PCR (RT-qPCR) analysis of *TaAnn12* in response to *Pst* and exogenous hormone treatments. (**a**) Accumulation of *TaAnn12* was significantly induced in incompatible interaction. Su11 leaves were inoculated with CYR23 (incompatible interaction) or CYR31 (compatible interaction). (**b**) The transcription of *TaAnn12* responded to exogenous hormone treatments. The 2^−ΔΔt^ method was used to calculate the relative expression of genes. Error bars represent ±SDs of three biological replications. Asterisks (*) indicate significant differences (*p* < 0.05) from 0 hpi.

**Figure 2 ijms-24-16381-f002:**
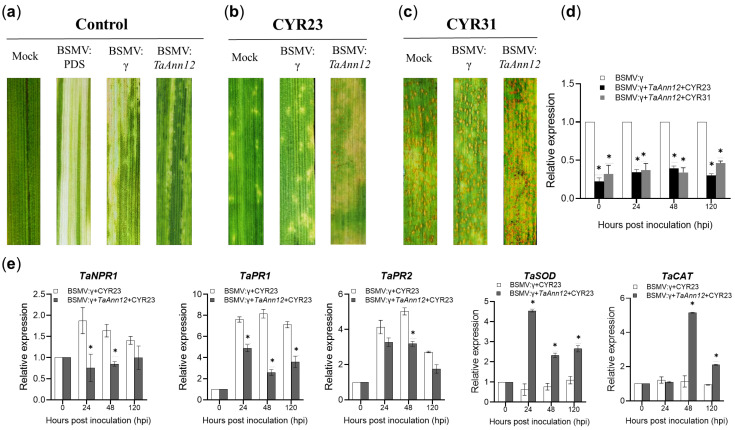
*TaAnn12* functions in response to *Pst* as indicated by the barley stripe mosaic virus (BSMV)-induced gene silencing. (**a**) At 10 d after inoculating with BSMV RNAs, the photo-bleaching and mild chlorotic mosaic symptoms were detected and photographed on wheat leaves. Wheat leaves were inoculated with *Pst* races CYR23 (**b**) and CYR31 (**c**), and the typical leaves were photographed at 15 dpi. (**d**) The expression level of *TaAnn12*. (**e**) Relative expression levels of resistance-related genes in *TaAnn12*-silenced leaves inoculated with CYR23. *TaNPR1*: non-expression of PR1, *TaPR1*: pathogenesis-related protein, *TaPR2*: beta-1,3-glucanase, *TaSOD*: superoxide dismutase, *TaCAT*: catalase. Error bars represent ±SDs of three biological replications. Asterisks (*) indicate significant differences (*p* < 0.05).

**Figure 3 ijms-24-16381-f003:**
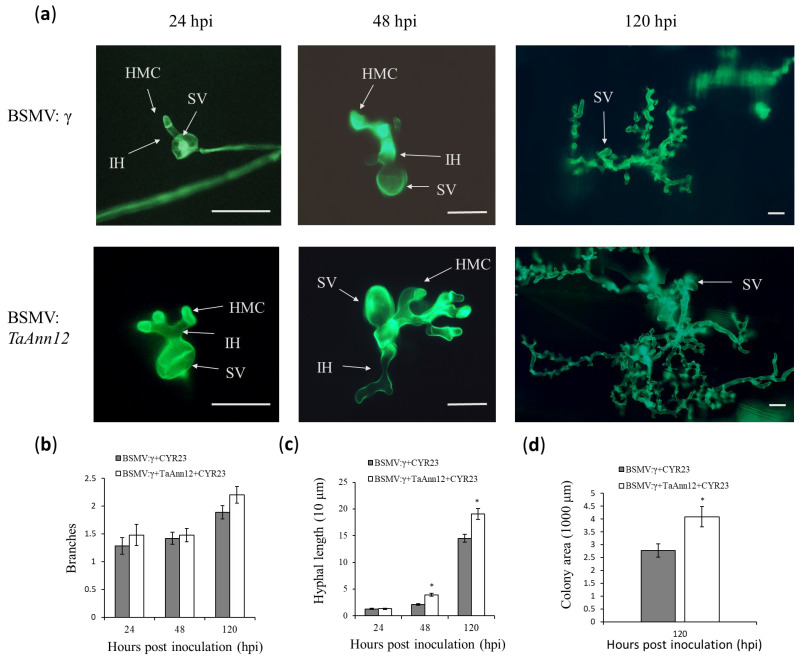
Histological observation of *Pst* race CYR23 growth in *TaAnn12*-silenced plants. (**a**) Expansion of stripe rust was observed under a fluorescence microscope. HMC, haustorial mother cell; IH, infection hypha; SV, substomatal vesicle. Bar: 20 μm. Average branches (**b**), hyphal length (**c**), and colony area (**d**) were calculated at 30 infection sites. Asterisks (*) indicate significant differences (*p* < 0.05).

**Figure 4 ijms-24-16381-f004:**
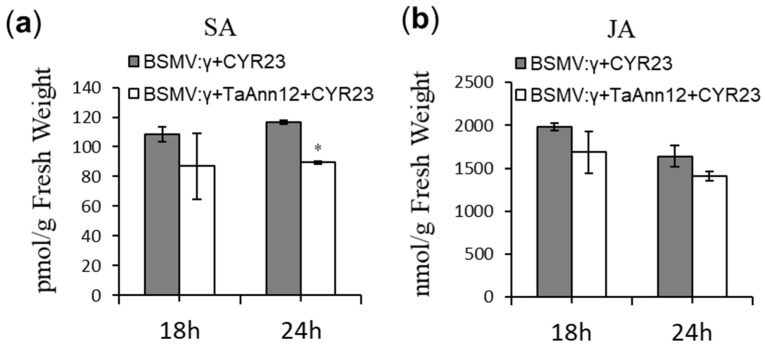
Accumulation of salicylic acid (**a**) and jasmonate (**b**) in *TaAnn12*-silenced plants after challenging with *Pst* CYR23. Error bars represent ± SDs of three biological replications. Asterisk (*) indicates a significant difference (*p* < 0.05).

**Figure 5 ijms-24-16381-f005:**
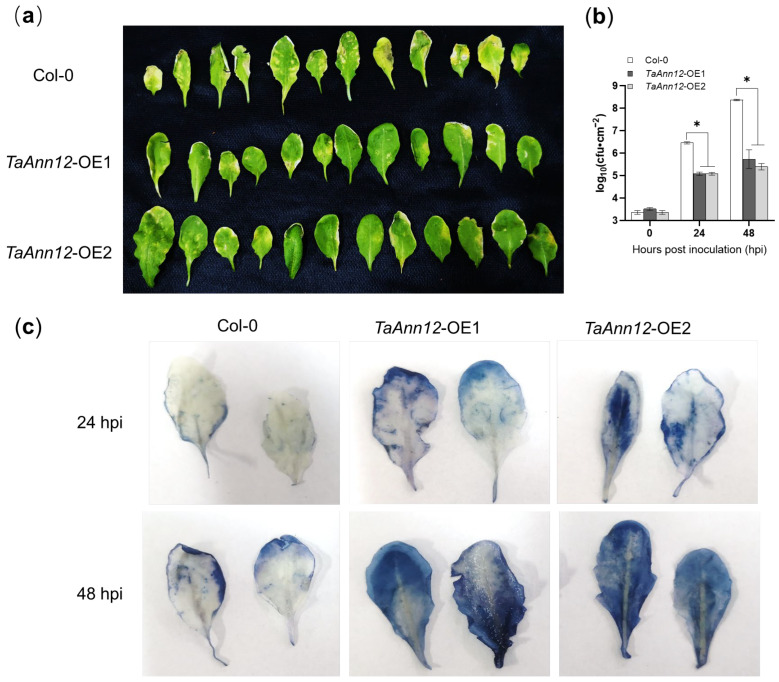
The phonotypes and cell death of *TaAnn12*-overexpressing lines inoculated with *Pto* DC3000. (**a**) Leaf images of Col-0, *TaAnn12*-T3-OE1, and *TaAnn12*-T3-OE2 at 4 d after *Pto* DC3000 inoculation. (**b**) Leaf colony density after *Pto* DC3000 incubation, where log10 represents the unit area cfu value after a base-10 logarithm. Asterisks (*) indicate significant differences (*p* < 0.05). (**c**) The cell death in *TaAnn12*-overexpressing lines induced by *Pto* DC3000.

**Figure 6 ijms-24-16381-f006:**
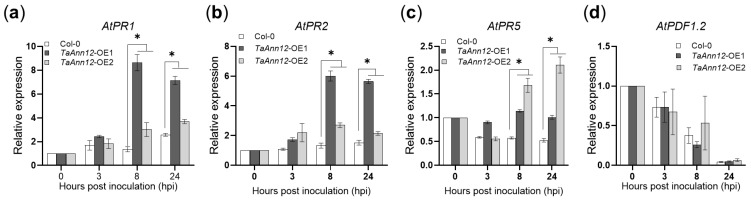
Relative expression levels of pathogenesis-related genes in *TaAnn12* transgenic plants after inoculating with *Pto* DC3000. (**a**) *AtPR1* pathogenesis-related protein. (**b**) *AtPR2* beta-1,3-glucanase. (**c**) *AtPR5* thaumatin-like protein. (**d**) *AtPDF1.2* plant defensin 1.2. Bars indicate the means ± SDs of three independent replicates. Asterisks (*) indicate significant differences (*p* < 0.05).

**Figure 7 ijms-24-16381-f007:**
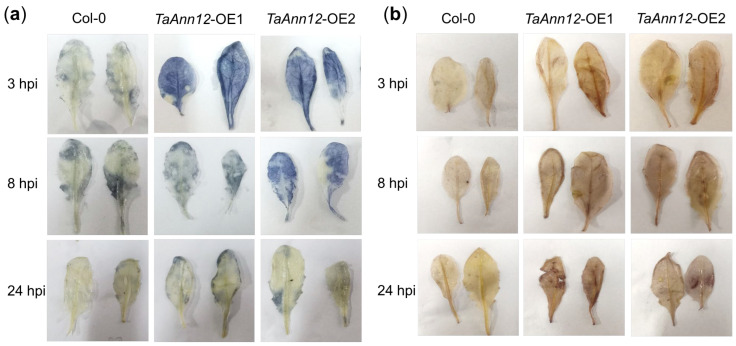
The accumulation of superoxide (**a**) and peroxide (**b**) in Arabidopsis leaves.

**Figure 8 ijms-24-16381-f008:**
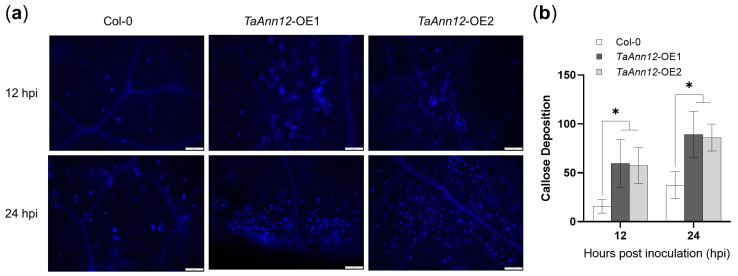
The callose deposition in *TaAnn12*-overexpressing lines induced by *Pto* DC3000. (**a**) Photographs of callose deposition in Arabidopsis leaves after inoculation with *Pto* DC3000. Leaves were stained with 0.01% aniline blue. Bars: 100 µm. (**b**) Quantification of callose deposition in Arabidopsis leaves induced by *Pto* DC3000. Asterisks (*) indicate significant differences (*p* < 0.05).

## Data Availability

Data supporting the reported results can be found in the relevant Appendix A.

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
