# Peer review of "The Wheat Annexin *TaAnn12* Plays Positive Roles in Plant Disease Resistance by Regulating the Accumulation of Reactive Oxygen Species and Callose"

_ijms, 2023, doi:10.3390/ijms242216381_

Round 1
Reviewer 1 Report
Comments and Suggestions for Authors
Please see the attachment.

Please review the text carefully, to eliminate spelling and grammatical errors.
Author Response
- The misspelled “callose” instead of “callose” was used, including in the manuscript title.
Response: This is our mistake. We have replaced the misspelled words.
- Analyzed in relation to the content of the Supplementary Material, the explanations regarding Figures SF.1 and SF.2, inserted in lines 408-410, are incomplete and confusing.
Response: Thanks for your suggestion. We added descriptions of Supplementary Materials.
Figure S1: Real-time quantitative PCR (RT-qPCR) analysis of TaAnn12 expression characteristics. (a) The transcription of TaAnn12 response to abiotic stress. (b) Different distribution of TaAnn12 in pivotal organs. Error bars represent standard errors from three biological replications. Error bars represent ± SD from three biological replications. Asterisks (*) indicate a significant difference (P < 0.05). Figure S2: Identification of TaAnn12 overexpression transgenic plants in T2 generations. Positive TaAnn12 transgenic plants were identified using PCR. (lines 421-425)
Reviewer 2 Report
Comments and Suggestions for Authors
The manuscript “The wheat annexin, TaAnn12, plays positive roles in plant disease resistance by regulating the accumulation of ROS and acllose” shows the studies done by the authors, to understand the role of one annexin from wheat on the resistance responses to Puccinia. The results obtained are interesting and the experiment design was consistent with the objectives.
Below are some issues that should be reviewed:
Introduction - (First paragraph) A better characterization of annexins should be done, as well as a better explanation of the increased interest in this group of plant proteins, since several evidences suggested their role in the regulation of the development and in the responses to a number of different environmental stimuli such as abiotic and biotic stresses.
Results and Material and Methods - The histological observations should include the quantification along the infection process of the haustoria both in BSMV: g and BSMV:TaAnn12. This will allow us to understand if the role of TaAnn12 is pre or post-haustorial as well as to know if the number of haustoria formation is increased in the BSMV:TaAnn12.
The presentation of Real time PCR is more robust if the authors use more than one reference gene, so I propose to include at least one more reference gene.
Minor issues
Title and abstract Calose not acllose
Line 54-56 The phrase should be clarified.
Line 66 Delete space.
Line 81-82 This sentence is one of the main conclusions of this manuscript, so it should be in the conclusions. In this last paragraph of the introduction, we expected to understand the main question as well as the approach design by the authors, not the main findings.
Figure 1 (b) The chart could be bigger to allow a better understanding of the results. The caption of this figure should include the methodology used for the determination of the relative expression.
Figure 2 (c) – The photograph of mock inoculation seems to show the appearance of uredia, the authors should review this.
Figure 2 (e) The chart should be bigger to allow a better understanding of the results.
Line 289 What is the meaning of (semi)biotroph?
